# Parental views on their children's smartphone use during personal and relational activities

**Kateřina Lukavská**[ID][1,2], **Roman Gabrhelík**[ID][1,2,3]*

**1** Department of Psychology, Faculty of Education, Charles University, Prague, Czech Republic,
**2** Department of Addictology, First Faculty of Medicine, Charles University, Prague, Czech Republic,
**3** Department of Addictology, General University Hospital in Prague, Prague, Czech Republic

* roman.gabrhelik@lf1.cuni.cz

## Abstract

Given the negative outcomes associated with smartphone use during personal and relational activities (SUPRA), parents strive to regulate its use among their children. However, media parenting recommendations lack knowledge of parental views on SUPRA and their relative occurrence in youths. This study aimed to (i) estimate SUPRA frequency among children and adolescents, (ii) assess parental dislike (PD) of SUPRA, and (iii) identify predictors of PD of SUPRA. An online survey was completed by 826 parents (49% mothers, age 25–74, Median = 43 years), capturing PD of SUPRA, estimated frequency of SUPRA in their children (49% female, age 6–18, Median = 10 years), parenting styles (warmth and control), parental attitudes toward screen media, and sociodemographic characteristics. The rate of frequent SUPRA was significantly higher in adolescents (2.7–48.1%) compared to children (2.1–27.2%) with odds ratios ranging from 0.67 to 3.04, depending on the activity type. PD of SUPRA was high in parents of children (M = 4.04, SD = 0.66) and adolescents (M = 3.93, SD = 0.71). Linear regression identified being a mother, having higher levels of control and warmth, and less positive attitudes toward screen media as significant predictors of SUPRA dislike. Our study was the first to report the estimated occurrence and parental dislike of smartphone use (SU) during various personal and relational activities, enabling their direct comparison. Experts warn against SU while studying and at bedtime, which frequently occurred in 4–5% of children and 10–12% of adolescents. More attention should be paid to SU during relational (peer/family) activities due to its relatively high occurrence and parental dissatisfaction.

## Introduction

Children and adolescents grow up surrounded by devices with electronic screens, such as televisions, computers, tablets and smartphones. Among elementary school-aged children, smartphones are the major source of screen time, contributing significantly to excessive screen use [1], which has been found to be related to adiposity (obesity) [2], lack of sleep and poor sleep quality [3–5], impairment of attention and other cognitive functions [6], and overall decrease in psychological well-being among children and adolescents [7].

**Data Availability Statement:** Data are available in a public repository: https://osf.io/tx8za/files/osfstorage/6650ab926b6c8e0e8b04caae.

**Funding:** This study was supported by Czech Science Foundation: 21-31474S and by Charles University: Cooperatio (Health Sciences).

**Competing interests:** The authors have declared that no competing interests exist.

More importantly, smartphones–being portable devices–can be used almost anywhere and at any time. The context of screen use (where, when, and why) has recently been proposed to be at least equally as important as its extent (i.e., screen time) [8] since the use of screens may interfere with important personal and social activities. For instance, smartphone use (SU) during interpersonal interactions, which has been also studied under terms "technoference" or "phubbing", seems to be associated with lower relationship satisfaction between friends [9] as well as between romantic partners [10–13]. In the professional context, SU during face-to-face negotiation led negotiators to appear less trustworthy and less professional [14].

Regarding family life, parental use during parent–child interactions (parental technoference) has been linked to decreased family satisfaction [15], suboptimal parent–child interactions, children's externalising and internalising behavioural problems [12], and parental inattentiveness to children's safety and emotional needs [16]. Although distractive SU during family interactions has been predominantly studied among parents, research has also shown that adolescents experience occasional (42%) or frequent (30%) distractions from their smartphones during face-to-face conversations with their parents [17].

Personal routines and activities may also be negatively affected by SU. In elementary school-aged children, the use of smartphones before or at bedtime was found to be associated with night-time awakenings and/or sleep disturbances [4]. Using smartphones while walking was shown to be associated with an increased risk of pedestrian accidents and injuries [18]. In teenagers, SU (WhatsApp) during learning led to decreased working memory performance and lower overall learning efficiency [19].

On the other hand, it has been argued that using smartphones also has benefits [20], plays an important role in adolescents' leisure time [21] and even may help to build enjoyable family leisure, stay connected with family members and increase a sense of belonging [22, 23]. Views on where, when and for what purposes SU is in/appropriate may vary from person to person, although some attitudes regarding SU seem to be quite universal, e.g., most parents believe that SU should be avoided during family mealtimes [24]. In addition, both parents and children seem to value balance between screen use and other activities (hobbies and chores) [25]. Little is known about parental views on SU during other specific personal, peer and family activities. However, a few studies have focused on parental views on their offspring's screen use in general. According to a relatively recent (but pre-covid) small-scale study, about 63% of parents of school-aged children reported that screen use within family somewhat affected their parent-child relationship either positively or negatively [26] and more than half of parents expressed to be concerned by screen use of their children [26]. Regarding gender differences, parents were found to be more concerned with their daughters' use of technology compared to their sons' [27]. This may be due to perceived differences in the purposes of technology use, where daughters are believed to use it predominantly for social networking and sons for gaming [27]. An older qualitative study revealed that most parents of adolescents expressed the high concern around their children smartphone use and its effect on their adolescents' mental health and behaviours (e.g., energy levels, ability to focus, time management or tendency to violence), physical health (e.g., eyesight, postures, bodily discomfort, physical activity level), social development (e.g., social skills, family bonding) and specific online risks (e.g., cyberbullying) [28].

## The role of parents in screen media regulation

Experts urge that screen media use in children should be regulated [29, 30] and that this responsibility relies mostly on parents [31]. The American Academy of Paediatrics (AAP) regularly releases recommendations for parents regarding the regulation of screen media, with

the latest update occurring in April 2024 [32]. AAP guidelines are probably the most widely recognized. Although they are relatively general concerning screen media use, some parts are specifically relevant to smartphone use, particularly the fourth component called 'Crowding Out,' which prompts parents to ask: 'What does media get in the way of? For parents of children with their own smartphone it is recommend to prompt children to restrain from device use during school, homework, bedtimes [32]. Similar guidelines have been proposed by the Italian Paediatric Society [1], recommending parents to restrict their children from using smartphones during mealtimes, bedtimes, and homework.

Current research, however, suggests that media parenting (active and restrictive parental mediation) has only a limited effect on children screen media use. Namely, restrictive practices are associated with somewhat lower screen media time in younger children [33] but do not seem to have a preventative effect against problematic screen media use [33–35], which can be defined as excessive media engagement with impaired control over use and with negative consequences in major areas of life functioning [36]. In older adolescents, restrictive parenting was positively and moderately associated with increased problematic online screen media use (the pooled correlation was found to be .25 [.05, .44]) [34]. This positive correlation between increased restrictive parenting and problematic media use could be partially explained by cross-sectional design of most studies, while it makes sense that when problematic use emerges, the restrictive activity of parents increases [34]. The similar pattern has been revealed using qualitative data [37]. However, longitudinal data suggested that problematic use led to increased restrictions but the restrictions further promoted problematic use in the next wave [38].

A recent study on gaming suggested that behind the promoting link between restrictive parenting and problematic media use (PMU) might be negative parental attitudes towards screen media [39]. Parents with negative attitudes were more prone to use restrictive-controlling approaches, which prompted children's defiance and were related to problematic gaming [39]. Qualitative data, which may offer more comprehensive overview, suggested that parents with greater knowledge, larger experience and with positive attitudes toward digital media were more controlling but also more participative, engaged and supportive [37].

The warm parent–child relationship seems to be a consistent negative correlate of PMU, both in younger and older adolescents [34]. High parental warmth (responsiveness) together with high parental control (demandingness) constitute an authoritative parenting style, which has also been investigated in relation to PMU. A cohort study of 14-year-old adolescents found the lowest prevalence of PMU (4%) in adolescents with authoritative parents and the highest prevalence of PMU (21%) in adolescents with authoritarian (low warmth and high control) mothers and neglectful (low warmth and low control) fathers, which suggested that the absence of parental warmth was associated with adolescent problematic screen media use [40].

The lack of parental warmth may be both the cause and the result of children's problematic screen media use. It has been argued that parent–child conflicts over (the extent of) SU may occur and are associated with decreased well-being and increased probability of suicide attempts among adolescents [41]. In summary, parents are prompted to regulate their children's screen media use, but they should try to maintain a positive attitude towards screen media and a positive relationship with their child to avoid sabotaging their regulative efforts. Smartphones may pose a significant challenge in this respect. Given the penetration of smartphones into various daily activities, including parent–child interactions, it may be very difficult for parents to regulate their use without initiating conflicts or developing frustration. Despite the importance of this issue for effective media parenting, research on parental perceptions of children's SU during personal, peer, and family activities (SUPRA) is scarce.

### This study

The aims of this study were (i) to estimate the frequency of SU during various personal (e.g., bedtime), and relational (e.g., family mealtimes, parent–child interaction, child's interaction with peers) activities (SUPRA) in children and adolescents based on their parents' views, and (ii) to analyse parental dislike (PD) of children's SUPRA in terms of their magnitude based on activity type and based on parental/child variables (e.g., parental warmth and control, parental attitudes towards screen media, parental gender, parental age, socioeconomic status–SES, child gender, child age, child smartphone time).

## Methods

### Data collection

This study was conducted in the Czech Republic. The participants were recruited (and data were collected) via an online panel operated by the Czech branch of The European National Panels. This company adheres to the ICC/ESOMAR International Code on Market, Opinion and Social Research and Data Analytics, which sets the standard of ethical and professional conduct for the global data, research, and insights community. The criteria for the inclusion of participants were being an adult (>18 years old) and being a parent of at least one child aged between 6 and 18 years. The target amount was 800 participants balanced for parental gender, child gender and child age. Data were collected 25–29 October 2022. The median time to complete the questionnaire was 11 minutes.

### Participants

A total of 1727 eligible persons were invited to participate, 880 agreed to participate and entered the online survey, and 836 participants completed the survey. Ten participants were excluded due to exceeding the quota. The final sample consisted of 826 participants. Parents with more than one child were asked to report on their youngest child within the eligible range (6–18 years). Participants were living in the household with one child (N = 466), two children (N = 308), or three or more children (N = 52). Participating mothers (49%) and fathers came from different families, and one participant represented each family. Two subsamples were established based on the age of the child for which a survey was completed: children aged 6–10 years (N = 423; 52.2% female) and adolescents aged 11–18 years (N = 403; 44.7% female). The characteristics of the samples are reported in S1 Table.

### Measures

**Smartphone Use during Personal and Relational Activities (SUPRA).** Fourteen personal and relational activities during which a smartphone can be used were proposed. Eight activities were derived from the scientific literature cited above which suggested that smartphone use during these activities might be harmful:

- When your child is supposed to be focusing on something else (e.g., studying) [Personal activity]

- At bedtime [Personal activity]

- During a conversation with a peer (e.g., sibling, friend) [Relational activity]

- When playing with a peer (e.g., sibling, friend) [Relational activity]

- When you are saying something important to your child [Relational activity]

- During a parent–child conversation [Relational activity]

- During a family mealtime [Relational activity]

- During dinning in the restaurant [Relational activity]

Six additional activities were included to achieve greater variety in both personal and family activities:

- While waiting (e.g., at bus stop) [Personal activity]

- In the bathroom (toilet) [Personal activity]

- When visiting family friends [Relational activity]

- When walking or hiking or being on trip together [Relational activity]

- When attending a cultural performance (e.g., theatre, cinema, concert) [Relational activity]

- When travelling/commuting together [Relational activity]

The participants were asked to estimate how often occurred each SUPRA in their offspring. The participants responded using the ordinal scale of Never/Sometimes/Frequently/I don't know, or I prefer not to say. The proportion of participants giving each type of response was computed for each SUPRA item. In addition, the frequent SUPRA score was calculated as the sum of items during which the frequent SU was reported. Cronbach's α of frequent SUPRA was 0.78, and McDonald's ω was 0.81.

**Parental dislike of children's smartphone use during personal and relational activities.** The same fourteen activities introduced above were used to assess PD of SUPRA.

The participants were asked to express how they would feel about their children's SU during these activities. The participants responded using a 5-point Likert scale ranging from "I would not mind it at all" (1) to "I couldn't stand it" (5). Mean values were computed for each item (i.e., activity). Additionally, the total score was calculated as the average of responses on individual SUPRA items. Parental dislike (PD) of SUPRA was treated as an interval variable. A higher score indicated a higher PD of SUPRA. Cronbach's α was 0.85, and McDonald's ω was 0.87.

**Parental Positive Attitudes Towards Screen activities (PATS).** PATS was assessed by ten items, which were constructed partially based on previous literature [39] and partially based on data from six focus groups with parents of elementary school-aged children in the Czech Republic between April and September 2021 (manuscript in preparation). The items reflected parental positivity, warmth and acceptance towards screen media (e.g., I totally understand why my child likes screens so much; I like to try new digital content and trends if my child introduces them to me; I find it annoying if my child spends time with his or her friend/s in front of screens instead of engaging in a non-screen activity (reverse coded)). The participants responded using a 4-point Likert scale ranging from "I totally disagree" (1) to "I totally agree" (4). Cronbach's α and McDonald's ω were 0.74.

## Parental warmth

General parental warmth was measured using 8 items from the Parental Acceptance-Rejection/Control Questionnaire (PARQ/C) [42], which is based on the theory of Parental Acceptance-Rejection [43]. Parental warmth reflects the magnitude of parental responsiveness and affection towards the child through expressing interest and positive feelings towards the child, praising the child's opinion, etc. (e.g., "I make my child feel that what (s)he does is important."). Participants responded to each item using ordinal scale of Almost Never True / Rarely True / Sometimes True / Almost Always True. Cronbach's α and McDonald's ω were 0.90.

## Parental control

General parental control was measured using 12 items from PARQ/C [42] describing parental regulative behaviour, such as monitoring children's whereabouts and activities, setting rules, and limiting children's freedom (e.g., "I tell my child exactly what time to be home when (s)he goes out."). Cronbach's α was 0.73, and McDonald's ω was 0.76. Participants responded to each item using ordinal scale of Almost Never True / Rarely True / Sometimes True / Almost Always True.

## Child's ownership of screen-based devices

The participants reported whether their child had his or her own smartphone, tablet, gaming console, computer, or television. The participants responded "yes" or "no" for each device. We asked about the child's personal ownership of devices, i.e., whether a child had a device for her exclusive use. This variable did not concern devices used by multiple members of a family.

## Smartphone time

The participants estimated the amount of time (in minutes) that their children usually spent with a smartphone during an average day.

## Sociodemographic characteristics of parents, children and families (Households)

Standard characteristics such as the age and gender of both parents and children, region, urban/rural area of residence, etc., were collected. The *socioeconomic status (SES)* of the household was measured using the abcde classification developed by Nielsen Admosphere [44] based on educational attainment, occupational status, region, and property. Abcde categorisation was developed to classify households into eight ordinal categories (A, B, C1, C2, C3, D1, D2, E) with an estimated population prevalence of 12.5% for each category. Given the lower prevalence of D1, D2 and E in our sample, we condensed these three categories into one and therefore had six ordinal categories.

## Statistical analysis

Analyses were conducted separately for parents of children (6–10 years old) and parents of adolescents (11–18 years old). Two categories were formed based on the presumed differences in the screen media use of (and the recommended parenting strategies for) children younger and older than 10 years [32].

Frequency analyses were conducted to show the rate of children and adolescents who use a smartphone (never, sometimes or frequently) during a given activity according to parental report. The differences between the children and adolescents in the rate of frequent SUPRA were analysed using the chi-square test of associations with Odds ratios to assess the effect sizes. The differences between children and adolescents in the total number of frequent SUPRA were analysed using Welch's *t test* and Cohen's *d* to assess the effect size.

Average PD of SUPRA was computed for each individual personal/relational activity separately for parents of children and adolescents. To assess differences in PD of SUPRA between the two groups of parents, Welch *t test*s with Cohen's *d*s to measure effect sizes were used.

Sociodemographic and other potential correlates of PD on SUPRA (e.g., PATS, parental warmth and control, smartphone ownership, smartphone time) were analysed using Spearman correlations and analysis of variance. As the final step, a linear regression model was proposed to explain the variance in PD of SUPRA with PATS, parental warmth and control, and

parental gender as presumed predictors. These predictors were selected based on whether they were found to be significantly associated with PD of SUPRA, i.e., on previous step.

Analyses and visualisations were conducted in R [45].

### Ethics

The ethical committee of the Ethical Committee of the Faculty of Education, Charles University, Prague, Czech Republic (no. 11/2020) approved the study. All subjects were informed about the study, and all provided informed consent. The consent was acquired in written form, with human participants actively marking the checkbox. Only after completing this action they were allowed to proceed further.

## Results

### Smartphone use during personal and relational activities

In children aged 6–10 years, the occurrence of any SUPRA was found to be relatively low, especially during dining in a restaurant, attending a cultural performance, at bedtime, and during a family trip, when close to two-thirds of children never used a smartphone according to their parents in these situations (Table 1, Fig 1). More than 50% of children sometimes used a smartphone when interacting with peers and 41% when interacting with parents. Only a minority of children displayed frequent SUPRA This applied even to activities during which SU was not disliked by parents (commuting, waiting)–see above.

The rate of adolescents who frequently used smartphones during personal and relational activities was higher than that of children but still did not exceed 17% in most activities–i.e., except commuting, waiting and being in the bathroom (Table 1, Fig 1). Some activities, however, deserve a closer look. As many as 47% of adolescents sometimes used a smartphone during bedtime; and additional 10% did so frequently. Even more adolescents used a smartphone when they were supposed to be focusing on something else (e.g., studying): 51% sometimes and additional 12% frequently.

The differences in the rate of frequent SUPRA between the children and adolescents were significant in most items (Table 1). The most pronounced child–adolescent differences in the frequent use were found in items "When a child is supposed to be focusing on something else (e.g., studying)" and "When walking or hiking or being on trip together" with both ORs equal to 3 (Table 1).

### Parental views on smartphone use during personal and relational activities

PD of SUPRA was generally high. In parents of children (6–10 years), the proportion of participants who would mind or could not stand their child's SU was approximately 80% (71–93%) in the case of all activities except commuting (37%), waiting (20%) and being on toilet (45%)– S2 Table. Interestingly, parents disliked SU not only during parent–child/family activities (e.g., family mealtimes or family trips) but also during children's personal activities (e.g., during studying, during bedtime). Similar results were observed in parents of adolescents (S3 Table). The proportion of parents who disliked SUPRA was again high (66–88%) in all activities except commuting (35%), waiting (22%), and being on the toilet (37%). In both groups of parents, the highest PD was observed for following SUPRA: (i) during studying (93% of child group and 88% of adolescent group expressed dislike), (ii) during cultural performance attended by a family (93% of child group, 89% of adolescent group), (iii) when a parent is trying to say something important to his or her child (91% of child group, 87% of adolescent group), and (iv) during bedtime (90% of child group, 86% of adolescents group).

**Table 1. The frequencies of SUPRA in children 6–10 years old (N = 423; 63% with their own smartphone) and adolescents 11–18 years old (N = 403; 98% with their own smartphone) and the between-group differences in the proportions of frequent use (the differences in the distribution of "Frequently" was analysed for each SUPRA item).**

| How frequently occurs in your child | Children 6–10 years (N = 423) | | | Adolescents 11–18 years (N = 403) | | | Differences in frequent use | |
|---|---|---|---|---|---|---|---|---|
| | Never | Sometimes | Frequently | Never | Sometimes | Frequently | χ2 | Effect size |
| Smartphone Use during Relational Activities | | | | | | | | |
| During a family mealtime | 263 | 126 | 23 | 224 | 153 | 15 | 1.38 | OR = 0.672 |
| | 62.2% | 29.8% | 5.4% | 55.6% | 38.0% | 3.7% | p = .24 | (95% CI 0.35–1.31) |
| When visiting family friends | 168 | 219 | 22 | 99 | 255 | 37 | 4.93 | OR = 1.84 |
| | 39.7% | 51.8% | 5.2% | 24.6% | 63.3% | 9.2% | p = .026 | (95% CI 1.07–3.18) |
| During dinning in the restaurant | 285 | 115 | 13 | 217 | 151 | 27 | 5.89 | OR = 2.26 |
| | 67.4% | 27.2% | 3.1% | 53.8% | 37.5% | 6.7% | p = .015 | (95% CI 1.15–4.45) |
| When walking or hiking or being on trip together | 273 | 122 | 17 | 123 | 222 | 45 | 15.2 | OR = 3.00 |
| | 64.5% | 28.8% | 4.0% | 30.5% | 55.1% | 11.2% | p < .001 | (95% CI 1.69–5.34) |
| When travelling/commuting together | 99 | 200 | 115 | 39 | 161 | 194 | 38.7 | OR = 2.49 |
| | 23.4% | 47.3% | 27.2% | 9.7% | 40.0% | 48.1% | p < .001 | (95% CI 1.86–3.32) |
| When you are saying something important to your child | 253 | 148 | 15 | 208 | 157 | 27 | 4.25 | OR = 1.95 |
| | 59.8% | 35.0% | 3.5% | 51.6% | 39.0% | 6.7% | p = .039 | (95% CI 1.02–3.73) |
| During parent–child conversation | 225 | 173 | 17 | 179 | 187 | 24 | 1.64 | OR = 1.51 |
| | 53.2% | 40.9% | 4.0% | 44.4% | 46.4% | 6.0% | p = .20 | (95% CI 0.80–2.86) |
| During a conversation with his or her peer (e.g., sibling, friend etc.) | 128 | 234 | 42 | 64 | 246 | 67 | 8.08 | OR = 1.81 |
| | 30.3% | 55.3% | 9.9% | 15.9% | 61.0% | 16.6% | p = .004 | (95% CI 1.20–2.73) |
| When playing with a peer (e.g., sibling, friend etc.) | 152 | 219 | 35 | 82 | 230 | 65 | 12.0 | OR = 2.13 |
| | 35.9% | 51.8% | 8.3% | 20.3% | 57.1% | 16.1% | p < .001 | (95% CI 1.38–3.30) |
| When attending a cultural performance (e.g., theatre, cinema, concert) | 371 | 32 | 9 | 309 | 57 | 11 | 0.32 | OR = 1.29 |
| | 87.7% | 7.6% | 2.1% | 76.7% | 14.1% | 2.7% | p = .57 | (95% CI 0.53–3.15) |
| Smartphone Use during Personal Activities | | | | | | | | |
| When your child is supposed to focus on something else (e.g., studying) | 284 | 114 | 18 | 139 | 205 | 48 | 16.5 | OR = 3.04 |
| | 67.1% | 27.0% | 4.3% | 34.5% | 50.9% | 11.9% | p < .001 | (95% CI 1.74–5.33) |
| At bedtime | 274 | 118 | 21 | 156 | 190 | 40 | 7.43 | OR = 2.11 |
| | 64.8% | 27.9% | 5.0% | 38.7% | 47.1% | 9.9% | p = .006 | (95% CI 1.22–3.64) |
| While waiting (e.g., at bus stop) | 52 | 217 | 145 | 23 | 135 | 234 | 47.0 | OR = 2.65 |
| | 12.3% | 51.3% | 34.3% | 5.7% | 33.5% | 58.1% | p < .001 | (95% CI 2.00–3.52) |
| In the bathroom (toilet) | 202 | 136 | 66 | 76 | 142 | 136 | 36.8 | OR = 2.76 |
| | 47.8% | 32.2% | 15.6% | 18.9% | 35.2% | 33.7% | p < .001 | (95% CI 1.97–3.85) |
| Total number of frequent SUPRA items—Mean (SD) | | | 1.32 | | | 2.41 | t = 7.19 | Cohen d = 0.502 |
| | | | (1.88) | | | (2.42) | p < .001 | |

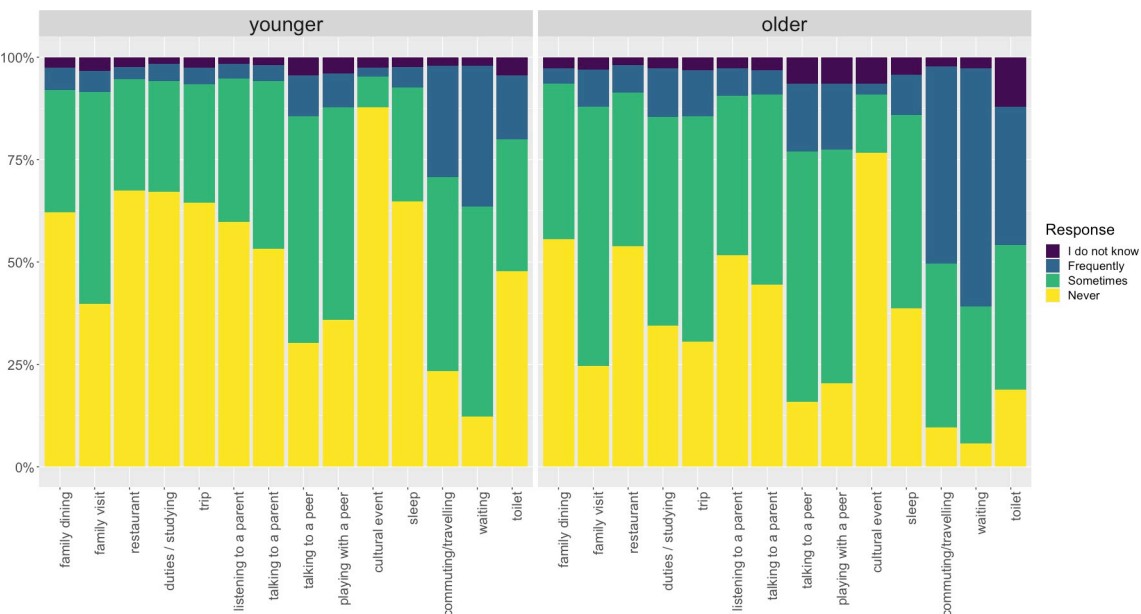

**Fig 1. Proportion of children (N = 423; 63% with their own smartphone) and adolescents (N = 403; 98% with their own smartphone) using smartphones never/sometimes/frequently during activities (based on parental report).**

To compare the two groups of parents, group means and standard deviations were calculated for each SUPRA item and for their averaged score. The parents of adolescents showed lower PD of SUPRA means in all fourteen items, but only in five of them were the differences statistically significant (Table 2). In only one routine–a family trip (walking or hiking or being on a family trip)–the difference was at least moderately strong. The responses showed surprisingly low between-subjects variability with standard deviations close to 1, suggesting a high level of agreement in PD of SUPRA among parents.

## Correlates of parental dislike of SUPRA

Parental dislike (PD) of SUPRA was negatively associated with SUPRA (Table 3) and with Parental positive attitudes toward screen activities (PATS). We found a small negative association between the PD of SUPRA and daily smartphone time but only in children aged 6–10 years. In both children and adolescents, we found a positive association between the PD of SUPRA and parental warmth and control (Table 3).

We found no significant association between PD of SUPRA and the collected sociodemographic characteristics except parental gender. Significantly higher dislike of SUPRA was found in woman-mothers (M = 4.09, SD = 0.674) than in men-fathers (M = 3.88, SD = 0.682): $t(824) = 4.41$, $p < .001$, Cohen $d = 0.31$.

## Predictors of parental dislike of SUPRA

Linear regression analysis showed that PATS, parental warmth and control, and parental gender all remained significant predictors of PD of SUPRA when assessed together in the regression model (Table 4). Each proposed variable significantly increased the explanatory power of the model (to the final 23.9% of explained variance). Using the principle of parsimony, we included only variables significantly associated with the PD of SUPRA in the model.

**Table 2. The comparison of parental dislike (PD) of SUPRA between parents of children 6–10 years old (N = 423) and parents of adolescents 11–18 years old (N = 403).**

| | Parents of children | Parents of adolescents | Differences | |
|---|---|---|---|---|
| | Mean (SD) | Mean (SD) | t | Effect size |
| Parental dislike of | | | | |
| *Smartphone Use during Relational Activities* | | | | |
| During a family mealtime | 4.34 | 4.33 | 0.173 | d = 0.012 |
| | (1.07) | (1.07) | p = .863 | |
| When visiting family friends | 4.01 | 3.95 | 0.814 | d = 0.057 |
| | (1.09) | (1.08) | p = .416 | |
| While dining in a restaurant | 4.40 | 4.28 | 1.653 | d = 0.115 |
| | (0.99) | (1.04) | p = .099 | |
| While walking or hiking or being on a trip together | 4.21 | 3.81 | 4.978 | d = 0.347 |
| | (1.13) | (1.17) | p < .001 | |
| While travelling/commuting together | 2.84 | 2.76 | 0.911 | d = 0.063 |
| | (1.36) | (1.32) | p = .363 | |
| When you are saying something important to your child | 4.65 | 4.54 | 1.838 | d = 0.128 |
| | (0.78) | (0.89) | p = .066 | |
| During a parent–child conversation | 4.41 | 4.28 | 1.891 | d = 0.132 |
| | (0.93) | (1.02) | p = .059 | |
| During a conversation with a peer (e.g., sibling, friend) | 3.96 | 3.72 | 3.140 | d = 0.22 |
| | (1.07) | (1.07) | p = .002 | |
| While playing with a peer (e.g., sibling, friend) | 3.91 | 3.78 | 1.693 | d = 0.118 |
| | (1.12) | (1.06) | p = .091 | |
| While attending a cultural event (e.g., theatre, movie, concert) | 4.71 | 4.63 | 1.462 | |
| | (0.76) | (0.83) | p = .144 | d = 0.102 |
| *Smartphone Use during Personal Activities* | | | | |
| When your child is supposed to be focusing on something else (e.g., studying) | 4.70 | 4.53 | 2.929 | d = 0.204 |
| | (0.73) | (0.89) | p = .004 | |
| At bedtime | 4.59 | 4.41 | 2.847 | d = 0.198 |
| | (0.87) | (0.96) | p = .005 | |
| While waiting (e.g., at bus stop) | 2.25 | 2.21 | 0.501 | d = 0.035 |
| | (1.21) | (1.21) | p = 0.616 | |
| Using the bathroom (toilet) | 3.19 | 2.88 | 3.363 | d = 0.234 |
| | (1.32) | (1.31) | p < .001 | |
| Total score–Mean (SD) | 4.04 | 3.93 | 2.355 | d = 0.164 |
| | (0.66) | (0.71) | p = .019 | |

## Discussion

Parents are prompted to regulate their children's screen media use. Guidelines concerning smartphones specifically have been provided by paediatricians and recommend that parents set rules restricting SU during mealtimes, during homework and during bedtime [1, 32]. These recommendations may reflect current evidence on the effects of SU but do not consider the perspective of parents on SU during these (and other) activities. This study attempted to bring forward parental perception on smartphone use during personal and relational activities (SUPRA) and the frequency of occurring SUPRA in their children. Our aim was to inform those who provide support of SU family regulation about potentially challenging types of SUPRA that would be worthy of further elaboration.

**Table 3. Spearman correlation coefficients between study variables in parents of children (N = 423) and parents of adolescents (N = 403).** Spearman ρ coefficients are presented.

| | | 1. | | 2. | | 3. | | 4. | | 5. |
|---|---|---|---|---|---|---|---|---|---|---|
| | Children (6–10 years) | | | | | | | | | |
| 1. | Parental dislike of SUPRA | - | | | | | | | | |
| 2. | SUPRA | -.308 | *** | - | | | | | | |
| 3. | PATS | -.246 | *** | .136 | ** | - | | | | |
| 4. | Parental warmth | .320 | *** | -.184 | *** | .003 | | - | | |
| 5. | Parental control | .253 | *** | -.112 | * | -.080 | | .248 | *** | - |
| 6. | Smartphone time | -.138 | ** | .337 | *** | .191 | *** | -.142 | ** | -.070 |
| | Adolescents (11–18 years) | | | | | | | | | |
| 1. | Parental dislike of SUPRA | - | | | | | | | | |
| 2. | SUPRA | -.154 | ** | - | | | | | | |
| 3. | PATS | -.214 | *** | -.146 | ** | - | | | | |
| 4. | Parental warmth | .244 | *** | -.205 | *** | .206 | *** | - | | |
| 5. | Parental control | .296 | *** | -.114 | * | -.033 | | .339 | *** | |
| 6. | Smartphone time | -.055 | | .257 | *** | -.128 | * | -.057 | | -.052 |

Note. PATS = parental positive attitudes towards child's screen use

SUPRA = smartphone use during personal and relational (peer/family) activities

As for the observed frequency of SUPRA, the substantial proportion of both children and adolescents showed at least occasional SU during social interactions, namely during interacting with peers (>65%) and with parents (>44%). Studies on technoference (or phubbing), i.e., SU during social interactions (e.g., texting someone else during a date or a conversation with a partner) showed that it is quite common behaviour also among adults [12], which however may negatively affect relationships [9]. As for the SU during parent-child interactions, the evidence on the harmful effect of parental technoference–i.e., from a parent to a child–has been provided [15, 46–48]. The effects of the opposite technoference (offspring technoference)–i.e., from a child to a parent–which has been reported by parents, are yet to be analysed by future studies. Both SU during peer-to-peer interactions and during parent-child interactions were disliked by parents. Neither SU during peer-to-peer interactions nor SU during parent-child interactions were not specifically addressed in current guidelines on SU regulation within family [1, 32].

Another important finding was the relatively high prevalence of at least occasional SU during bedtime among adolescents (57%). This is in congruence with previous findings of frequent SU during bedtime in 32% of adolescents [49], 44% of college students [50] and more than 80% of young adults [51]. All mentioned studies also argued that SU during bedtime

**Table 4. Coefficients for the linear regression model predicting PD of SUPRA ($R^2$ = 0.239), N = 826.**

| Predictor | Estimate | SE | t | P |
|---|---|---|---|---|
| PATS | -0.4024 | 0.0471 | -8.53 | < .001 |
| Parental warmth | 0.4715 | 0.0501 | 9.42 | < .001 |
| Parental control | 0.2668 | 0.0565 | 4.72 | < .001 |
| Parental gender: | | | | |
| Male | -0.0880 | 0.0428 | -2.06 | 0.040 |

Note. PATS = Parental Positive Attitudes Towards children's Screen use

negatively affected sleep [49–51], which has also been found by a recent systematic review [4]. The parental dislike of SU during bedtime was found to be very high, which might be explained by parental worries about the negative effects of SU on child's healthy development. The small scale study with parents of younger children (up to four years old) revealed that the significant proportion of parents (36%) found screen media use as harmful [52]. In parents of school-aged children some concern expressed approximately two thirds of parents [26]. And even the more frequently reported various concerns about screen use (and smartphone use specifically) parents of adolescents [28]. It is also worth mentioning that significant and substantial differences in SU during bedtime based on age were found by previous studies; the prevalence decreased with the increasing age of participants [51]. This generation gap may complicate the dialogue between parents and their children regarding SU during bedtime. It is probable that the involvement of the third party (educators, family/prevention professionals, psychologists) would be useful to mediate the dialogue and to provide arguments acceptable for both parties. No media use during bed time is given as the one of three example family rules concerning smartphones by paediatricians [1, 32].

Finally, occasional SU when a child is supposed to be focusing on something else (e.g., studying, doing homework) was found to be common (according to parents), which is in congruence with the high frequency reported by adolescents themselves [53]. At the same time, parental dislike of SU during this particular type of activity was very high. This is somewhat congruent with previous study with parents of younger children (up to four years old) which found that parents' major concern was the harmful effects of screen media use on their child's attention [52] and with small-scale study with parents of Australian teenagers who perceived screen devices as highly distractive even when their use was primarily motivated by learning goals [54] and as able to compromise their child's ability to deeply engage with tasks (e.g., homework) [55]. The very high parental dislike was reported for children SU when parents is trying to say something important to them, which is quite understandable as such SU combines technoference (phubbing)–see above–with the inability to focus attention due to SU. Among other SUPRA, that were very highly disliked by parents, but only marginally occurring, was SU when attending a cultural performance, which is also combination of relational and attention-requiring activity. The dislike in this case may be explained by parental wish to bond with their child and spend quality time together (relational part), which requires shared attention, that SU hinders. Despite these challenges, there are encouraging aspects. SU during attention-demanding activities (e.g., studying, doing homework) could be open to parental regulation, as adolescents themselves perceive it as distractive and are motivated to use some regulation strategies [53]. Parents could provide support, guidance and supervision in their children's self-regulative efforts.

To sum up, parental dislike (PD) of SUPRA was very high for (i) all relational activities, emphasizing the parental sensitivity to phubbing and reflecting worries about children's relationships, (ii) activities requiring deep attentional focus, which may result from parental worries about children school (or more broadly cognitive) performance, and finally (iii) use that may interfere with sleep, which may be explained by parental worries about child's daily functioning and healthy development. Contrary to that, most parents would not mind their child's SU during unoccupied time such as commuting or waiting in queue. It shows that parents are able to differentiate their views on SUPRA and that their dislike is rooted in some real concerns. More attention should be paid to discourses about screen media and parenting and how they shape parental attitudes towards screen media [56].

Less positive attitudes towards screen activities, higher parental warmth and control and being female (mother) were all found to significantly contribute to high PD of SUPRA even when analysed together using regression analysis. Positive parental attitudes towards screen

media have been previously studied mostly as a factor contributing to higher screen time in children, often alongside higher parental engagement in their own screen media use [57, 58]. This could suggest that positive parental attitudes are a risk factor for adolescent healthy SU. Negative parental attitudes towards media, however, seem to undermine the efficiency of media parenting [38, 39]. In addition, parental attitudes towards modern screens (such as smartphones) have been found to be associated with the age and education of parents: older and less-educated parents are less warm towards them and less ready to guide their children in their use [59]. This suggests that the effects of parental attitudes towards screen media might be ambivalent and requires further research. PD of SUPRA was positively associated with general parenting, i.e., higher parental warmth and control meant more pronounced dislike of children SUPRA. It has been found that parental warmth and control tend to decrease as children grow [40, 60]; however, the decrease in the PD of SUPRA, which we observed between parents of children and parents of adolescents, was only small and insignificant. The previous research suggests that parents who are more invested in the rearing of their children are also more aware and sensitive to the potential negative effects of SUPRA [37], which may contribute to their higher dislike. Previous research showed that parents tend to perceive both the positive and the negative aspects of screen media [22, 23, 26], and that child's gender [27], ethnicity [56], the type of media, the age of children [57] and previous experience [37] plays important role. Experts should objectively inform parents about both positive and negative aspects of SU to prevent unnecessary anxiety in parents, especially in those who are highly focused on child rearing.

The rate of frequent SUPRA among children and adolescents can be considered low, which seems positive given the current evidence on the negative effects of SU during interpersonal interactions [9, 11–13, 15, 19, 61] and personal routines [1, 4, 18, 19, 51]. This relatively low frequency of SUPRA might stem from the fairly high dislike of SUPRA among parents. This explanation seems to be supported by the fact that SU during bedtime was found to be more frequent among young adults (compared to adolescents) although there was a strong negative association between the frequency of SU during bedtime and age [51], suggesting that parental influence can decrease the frequency of SU during bedtime in adolescents. Notably, the magnitude of the association between the frequency of SUPRA and parental dislike was substantially higher in children than in adolescents. This may suggest a decrease in parental influence over SU as children grow older (and become adolescents). This is congruent with previous findings of decreased efficiency of media parenting on (problematic) media use in (older) adolescents compared to that in younger adolescents [33, 34] and also with qualitative study showing the emergence of conflict between parental need for authority and child's need for autonomy in using screen media among pre-adolescents [25]. SUPRA could be an important source of parental frustration, as parents of adolescents' dislike SUPRA but witness its increasing frequency as their children grow up. Prospective studies are warranted to observe these phenomena in their development, as we could not draw conclusions based on cross-sectional data. It should also be noted that our findings relied on parental reports, and as such, they might underestimate (or overestimate) the real frequency of SUPRA in children and adolescents. Further research assessing the frequency by children's and adolescents' reports is warranted to observe the parental influence over SUPRA more precisely.

Studies focusing on SU from the adolescent perspective have argued that smartphones constitute the central aspect of adolescent leisure time [21, 25]. The presence of a generation gap between parents and children concerning SUPRA seems to be probable, as the effects of age have been observed both for attitudes towards smartphones [59] and for actual behaviour [51]. The different views on SU between parents and their adolescent children has been already documented by a small-scale qualitative study, where parents reported to be much more

concerned about the harmful effects of SU compared to children [28]. As parent–child conflicts over SU can have serious consequences [41] and might negatively affect the efficiency of parental regulation of screen media use [39], it seems very important to mediate the dialogue between parents and children and help them reach agreements on SU during various personal and relational routines. Providing guidelines for parental regulation does not seem to be enough. The mediators should reflect the up-to-date evidence on the effects of SUPRA and the parental and child/adolescent perspectives on SUPRA. Currently, it is not easy to establish when and to what extent smartphones are relatively harmful or harmless to use for children and adolescents. To our knowledge, only SU while walking, while studying and during bedtime has been adequately scientifically investigated. It seems meaningful to restrict SU during these personal activities. Apart from these, SU during peer-to-peer and parent-child interactions should be addressed. SU during other activities (e.g., family mealtimes) should also be debated within families to reach an acceptable compromise for all members. Rules setting process should be cooperated by all family members and mediated by family experts as it seems that various barriers–including parental fear from conflicts with a child–exist to prevent parents from successful regulation [62].

This study has strengths. Studies on parental views on SUPRA are lacking, as are studies on child and adolescent SUPRA itself. This study fills this knowledge gap. We used a well-constructed sample of parents that allowed us to study phenomena separately among parents of preadolescent children aged 6–10 years and parents of adolescents aged 11–18 years. Our sample also included 50% fathers, which is an unusually high proportion, especially compared to convenience samples.

This study also has limitations. The participants were instructed to complete the questionnaire about their youngest child (6–18 years), which resulted in a high proportion of children with older siblings in the sample. Differences between the children and adolescents were analysed using cross-sectional between-subject comparisons, and therefore, these could reflect cohorts' specifics rather than developmental characteristics. The measurement did not distinguish for what purposes the smartphones were used by the children and adolescents (i.e., active or passive consumption of media), although we know that this might be an important factor affecting whether SU was beneficial or harmful [63] and could make an important difference in parental views on SUR. However, we specified that we were asking about SU, which was unrelated to the routine/activity in question (e.g., using a smartphone in a restaurant to read the menu on the web would not qualify as SUPRA). The findings on SUPRA frequency were based on parental reports, and as such, they might be under- or overestimated. Neither children's reports nor objective measures, such as technical screentime monitoring, were used. However, the primary focus of the study was parental views on SUPRA, not SUPRA itself. Employing such measures is recommended in future studies primarily targeting SU.

## Conclusion

Our study investigated parental perceptions of their children's smartphone use during personal and relational activities (SUPRA). The findings revealed that most parents expressed dislike of children's SU in various personal contexts, such as studying or bedtime, as well as during relational (parent-child and peer-to-peer interactions) and family-oriented activities like mealtimes or trips. This suggests that parents might be highly motivated to regulate various SUPRA in their children, which is an important message for family professionals. Another important message is that parents differentiated between activities in terms of not minding their children using smartphones during waiting or commuting, while being very much opposed to SU at bedtime or while studying. In several instances, parents observed their

children engaging in SUPRA. Our study suggests that families could benefit from professional guidance in addressing smartphone use. Further research is needed to explore SUPRA from the viewpoint of children and adolescents, thereby providing a comprehensive understanding of this topic.

## Supporting information

**S1 Table. Characteristics of participants.**
(DOCX)

**S2 Table. Parental views on children's smartphone use during personal and relational activities in parents of children aged 6–10 years (N = 423).** Frequencies of responses and their proportions in percentages are displayed.
(DOCX)

**S3 Table. Parental views on children's smartphone use during personal and relational activities in parents of adolescents aged 11–18 years (N = 403).** Frequencies of responses and their proportions in percentages are displayed.
(DOCX)

## Author Contributions

**Conceptualization:** Kateřina Lukavská, Roman Gabrhelík.

**Data curation:** Kateřina Lukavská.

**Formal analysis:** Kateřina Lukavská.

**Funding acquisition:** Kateřina Lukavská.

**Investigation:** Kateřina Lukavská, Roman Gabrhelík.

**Methodology:** Kateřina Lukavská.

**Project administration:** Kateřina Lukavská.

**Resources:** Kateřina Lukavská.

**Supervision:** Roman Gabrhelík.

**Visualization:** Kateřina Lukavská.

**Writing – original draft:** Kateřina Lukavská.

**Writing – review & editing:** Roman Gabrhelík.

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
