## [Decision Letter · Decision Letter 0]

10 Apr 2024

PONE-D-24-02065Parental views on their children’s smartphone use during personal and family routinesPLOS ONE

Dear Dr. Gabrhelík,

Thank you for submitting your manuscript to PLOS ONE. After careful consideration, we feel that it has merit but does not fully meet PLOS ONE’s publication criteria as it currently stands. Therefore, we invite you to submit a revised version of the manuscript that addresses the points raised during the review process.

Your manuscript has been evaluated by two reviewers and their comments are available below. They believe the manuscript would benefit from a more thorough explanation of the methodological details and justification of statistical models applied. Could you please revise your manuscript carefully to address all the comments?

We look forward to receiving your revised manuscript.

Kind regards,

Annesha Sil, Ph.D.

Associate Editor, PLOS ONE

Journal Requirements:

"This study was supported by Czech Science Foundation: 21-31474S and by Charles University: Cooperatio (Health Sciences)."

4. In the online submission form, you indicated that [The data underlying the results presented in the study are available from the corresponding author.]. 

Reviewers' comments:

Reviewer's Responses to Questions

**Comments to the Author**

1. Is the manuscript technically sound, and do the data support the conclusions?

Reviewer #1: Yes

Reviewer #2: Yes

2. Has the statistical analysis been performed appropriately and rigorously? 

Reviewer #1: Yes

Reviewer #2: No

3. Have the authors made all data underlying the findings in their manuscript fully available?

Reviewer #1: Yes

Reviewer #2: Yes

4. Is the manuscript presented in an intelligible fashion and written in standard English?

Reviewer #1: Yes

Reviewer #2: Yes

5. Review Comments to the Author

Reviewer #1: The paper "Parental views on their children’s smartphone use during personal and family routines" examines parental views on children's smartphone use during their daily routines and children's SU during these routines. The manuscript is well written, and the analyses are appropriate, but I have some issues with the terminology and the construct/definition of these routines.

For me, daily routines represent some mundane and "automatic" activities, such as brushing teeth, so I was confused when I read your manuscript. When looking into your items, they seem to be family/peer interactions or activities, not necessarily routines. I would recommend renaming these routines in such a way or providing a more clear explanation in the introduction of how and why these interactions and activities are considered routines.

I would also recommend dividing routines into categories of family and personal routines, as parents may have different views on these two categories and children may behave differently. Following this, I think abbreviations in the text would benefit from this (e.g., SUPR: personal routines, SUFR: family routines).

As I mentioned, some of the routines are related to interactions with family members or peers, and you look into how they are related to the SU. This can be seen as technoference and/or phubbing, and I think your paper would benefit from discussing your results in this context, at least for the routines that are related to the interaction with others.

All in all, this paper is novel in the field and shows some new perspectives on children's SU, but would benefit from more clear construct definition and clarification.

I would recommend not using abbreviations in the first sentence of the abstract. It would be better to write down the whole expression and put the abbreviation in the brackets (as is in the main text).

When describing scales of parental warmth and control, use the full name of the scale when first mentioning it.

There are also several citing issues and errors I saw throughout the manuscript; read and revise these errors carefully.

Reviewer #2: This study uses a sample of parents of children (aged 6-18) from Czech Republic to quantitatively describe parents’ perceptions of children’s use of smartphones during some routine events. The results reveal differences based on children’s age and some interesting associations with parents’ attitudes and parental warmth and control. Below are some of my suggestions for improvement:

1. In the abstract, it might be helpful to mention the age range and gender of both the parent sample and of their children.

2. Page 4, “Guidelines for parental regulation of specifically SU have been proposed by the Italian Paediatric Society,…”: What about from other organizations such as the American Academy of Pediatrics?

3. “this relationship could not be fully attributed to the cross-sectional design of synthesised studies [27].” I have a hard time understanding this sentence. Please clarify more.

4. Page 5: This is a really long paragraph. Please consider breaking it down.

5. Overall, in the introduction, there is a lack of in-depth literature review on children’s SUR and related phenomena, such as smartphone use during dinner time and during time spent with family. And there is a lack of literature review on parents’ perceptions of this.

6. A concerning part of the method is that the selection and definition of the 14 routines being studied are unclear and seem arbitrary. Please explain the scientific method/procedure for how these 14 routines were determined.

7. Some of the routines are quite confusing. For example, what does "during a visit" mean/?

8. For between-group comparison, why choosing 10/11 to be the breaking point for children versus adolescents?

9. I think results from the regression model should drive the main result interpretation and discussion rather than the correlations. In the regression model, it is important that many potential confounding factors, including the sociodemographic characteristics of both parents and children, are controlled.

6. PLOS authors have the option to publish the peer review history of their article (what does this mean?). If published, this will include your full peer review and any attached files.

Reviewer #1: No

Reviewer #2: No

---

## [Author Response · Author response to Decision Letter 0]

30 May 2024

Response to reviewers

Dear Editor, dear Reviewers,

We would like to thank the Editor and the Reviewers for their valuable comments on our manuscript. 

We believe that we have addressed all key concerns while preserving the scientific aims of the original submission. It is our opinion that the revised manuscript has improved in terms of quality and clarity. 

A detailed item-by-item responses to all comments follow. 

We have addressed all the additional Journal Requirements, incl. data availability and funding information.

Reviewer #1: 

The paper "Parental views on their children’s smartphone use during personal and family routines" examines parental views on children's smartphone use during their daily routines and children's SU during these routines. The manuscript is well written, and the analyses are appropriate, but I have some issues with the terminology and the construct/definition of these routines.

#1) For me, daily routines represent some mundane and "automatic" activities, such as brushing teeth, so I was confused when I read your manuscript. When looking into your items, they seem to be family/peer interactions or activities, not necessarily routines. I would recommend renaming these routines in such a way or providing a more clear explanation in the introduction of how and why these interactions and activities are considered routines. I would also recommend dividing routines into categories of family and personal routines, as parents may have different views on these two categories and children may behave differently. Following this, I think abbreviations in the text would benefit from this (e.g., SUPR: personal routines, SUFR: family routines).

Response: Thank you for your comment! We initially thought 'activity' was too general and chose 'routines' to emphasize the frequency and repetition of these activities. However, we now agree that 'activity' is more appropriate. We changed smartphone use during routines (SUR) to smartphone use during personal and relational activities (SUPRA). We specified more this while introducing the variable. For each SUPRA item we also specified whether it concerns personal or interpersonal/family activity. When reporting, we first reported all relational SUPRA and then all personal SUPRA items for higher clarity.

#2) As I mentioned, some of the routines are related to interactions with family members or peers, and you look into how they are related to the SU. This can be seen as technoference and/or phubbing, and I think your paper would benefit from discussing your results in this context, at least for the routines that are related to the interaction with others.

Response: Thank you for this suggestion. In the introduction, we clarified that the use of smartphones during interpersonal interactions has been previously studied under terms technoference/phubbing:

“For instance, smartphone use (SU) during interpersonal interactions, which has been previously studied under terms “technoference” or “phubbing”, seems to be associated with lower relationship satisfaction between friends [9] as well as between romantic partners [10–13]. In the professional context, smartphone use during face-to-face negotiation led negotiators to appear less trustworthy and less professional [14]. Regarding family life, parental use during parent–child interactions (parental technoference) has been linked to decreased family satisfaction [15], suboptimal parent–child interactions, children’s externalising and internalising behavioural problems [12], and parental inattentiveness to children’s safety and emotional needs [16]. SU during interpersonal interactions has been predominantly studied among adults, but the occurrence of occasional (42%) or frequent (30%) distraction by smartphones has also been reported for adolescent children during face-to-face conversations with their parents [17].”

#3) All in all, this paper is novel in the field and shows some new perspectives on children's SU, but would benefit from more clear construct definition and clarification.

Response: 

We clarified more the process how SUPRA was constructed:

“Fourteen personal, interpersonal and family activities during which a smartphone can be used were proposed. Eight activities were derived from the scientific literature cited above, which suggested that smartphone use during these activities might be harmful:

- When your child is supposed to be focusing on something else (e.g., studying) [Personal activity]

- At bedtime [Personal activity]

- During a conversation with a peer (e.g., sibling, friend) [Relational activity]

- When playing with a peer (e.g., sibling, friend) [Relational activity]

- When you are saying something important to your child [Relational activity]

- During a parent‒child conversation [Relational activity]

- During a family mealtime [Relational activity]

- During dinning in the restaurant [Relational activity]

Six additional activities were included to achieve greater variety in both personal and family activities:

- While waiting (e.g., at bus stop) [Personal activity]

- In the bathroom (toilet) [Personal activity]

- When visiting family friends [Relational activity]

- When walking or hiking or being on trip together [Relational activity]

- When attending a cultural performance (e.g., theatre, cinema, concert) [Relational activity]

- When travelling/commuting together [Relational activity]

We believe that this also helped us to better convey the message why to focus on smartphone use during personal and relational activities.

#4) I would recommend not using abbreviations in the first sentence of the abstract. It would be better to write down the whole expression and put the abbreviation in the brackets (as is in the main text).

Response: Thank you, changed accordingly.

#5) When describing scales of parental warmth and control, use the full name of the scale when first mentioning it.

Response: Thank you for your very appropriate suggestion. We have now included the full names of the scales when first mentioning the scales of parental warmth and control.

#6) There are also several citing issues and errors I saw throughout the manuscript; read and revise these errors carefully.

Response: Thank you, corrected.

-----------

Reviewer #2: 

This study uses a sample of parents of children (aged 6-18) from Czech Republic to quantitatively describe parents’ perceptions of children’s use of smartphones during some routine events. The results reveal differences based on children’s age and some interesting associations with parents’ attitudes and parental warmth and control. Below are some of my suggestions for improvement:

1. In the abstract, it might be helpful to mention the age range and gender of both the parent sample and of their children.

Response: Thank you. We have now included the age range and gender of both the parent sample and their children in the abstract.

2. Page 4, “Guidelines for parental regulation of specifically SU have been proposed by the Italian Paediatric Society,…”: What about from other organizations such as the American Academy of Pediatrics?

Response: Thank you for this comment. We reported guidelines proposed by Bozzola et al. (2019), because they specifically mentioned smartphone use (and its possible collision with important personal activities). In contrast, the AAP guidelines (COUNCIL ON COMMUNICATIONS AND MEDIA, 2016), while more widely recognized, , were older and rather general. However, as the updated AAP guidelines from April 2024 (American Academy of Pediatrics, 2024) addressed specifically smartphone use, we added them to the text:

“American Academy of Pediatrics (AAP) regularly publishes recommendations for parents concerning the regulation of screen media in general, with the newest update in April 2024 [26]. AAP guidelines are probably the most widely recognized. Although they are relatively general concerning screen media use, some parts are specifically relevant to smartphone use, particularly the fourth component called 'Crowding Out,' which prompts parents to ask: 'What does media get in the way of?'. For parents of children with their own smartphone it is recommend to prompt children to restrain from device use during school, homework, bedtimes [31]. Similar guidelines have been proposed by the Italian Paediatric Society [1], recommending parents to restrict their children from using smartphones during mealtimes, bedtimes, and homework.”

We are also aware that other societies (e.g., Canadian Paediatric Society, Institute de France Académie des Sciences, British Royal College for Pediatrics and Child Health) developed guidelines on screen media use, but as our focus was on smartphone use, which is only marginally mentioned or not addressed in these guidelines, we chose not to report on them in detail..

3. “this relationship could not be fully attributed to the cross-sectional design of synthesised studies [27].” I have a hard time understanding this sentence. Please clarify more.

Response: Thank you for your feedback. We have divided and rewritten the sentence for better clarity:

“In older adolescents, restrictive parenting was positively and moderately associated with increased problematic online screen media use (the pooled correlation was found to be .25 [.05, .44]) [29]. This positive correlation between increased restrictive parenting and problematic media use could be partially explained by cross-sectional design of most studies, while it makes sense that when problematic use emerges, the restrictive activity of parents increases [29]. However, a longitudinal study showed that the association was present even when the analysis was controlled for previous problematic use – i.e., problematic use led to increased restrictions but the restrictions further promoted problematic use in the next wave [31].”

4. Page 5: This is a really long paragraph. Please consider breaking it down.

Response: Thank you for your suggestion. We have reviewed the paragraph and have broken it down into shorter sections for better readability.

5. Overall, in the introduction, there is a lack of in-depth literature review on children’s SUR and related phenomena, such as smartphone use during dinner time and during time spent with family. And there is a lack of literature review on parents’ perceptions of this.

Response: We conducted a thorough literature search but identified only a small number of directly relevant sources. However, we provided a review on a closely related topic—parental views on screen media in general. In total, we have added 10 new sources.

6. A concerning part of the method is that the selection and definition of the 14 routines being studied are unclear and seem arbitrary. Please explain the scientific method/procedure for how these 14 routines were determined.

Response: Some routines (or activities as we renamed according to reviewer’s 1 suggestion) were derived from scientific literature, particularly those personal, interpersonal, and family activities during which smartphone use might be harmful; namely bedtime, bathroom time, activities requiring attentional focus – learning, interacting with peers, parent-child interactions, family mealtimes. We built upon this knowledge to derive 8 activities:

Personal activities (2 items)

- When he or she is supposed to be focusing on something else (e.g., studying)

- At bedtime

Interpersonal activities (either peer-to-peer or parent-child) (4 items)

- During a conversation with a peer (e.g., sibling, friend)

- When playing with a peer (e.g., sibling, friend)

- When you are saying something important to your child

- During a parent‒child conversation

Family activities (2 items)

- During a family mealtime

- During dinning in the restaurant

Additionally, we included other relevant activities identified in a qualitative study (Hrabec et al., manuscript in preparation) recently conducted with parents, which focused on their children's smartphone use.

Personal activities (2 more items)

- While waiting (e.g., at bus stop)

- In the bathroom (toilet)

Family activities (4 more activities)

- When visiting family friends

- When walking or hiking or being on trip together

- When attending a cultural performance (e.g., theatre, cinema, concert)

- When travelling/commuting together

We clarified this in the manuscript. Please refer to the Measure section for the detailed description.

7. Some of the routines are quite confusing. For example, what does "during a visit" mean/?

Response: Thank you very much for pointing this out. The original Czech item in full was 'When visiting family friends,' meaning the situation where the whole family (or a parent with a child) meets family friends. This was the context we asked parents to report about. We have now included the entire item for clarity.

8. For between-group comparison, why choosing 10/11 to be the breaking point for children versus adolescents?

Response: The choice of 10/11 as the breaking point between children and adolescents was based on the presumption that significant changes in screen media use occur around this age. This is also reflected in different guidelines for children younger than 10 years and older, such as those from the American Academy of Pediatrics (2024). We have clarified this in the Methods section when reporting the statistical analysis:

Analyses were conducted separately for parents of children (6–10 years old) and parents of adolescents (11–18 years old). Two categories were formed based on the presumed differences in the screen media use of (and the recommended parenting strategies for) children younger and older than 10 years (American Academy of Pediatrics, 2024).

9. I think results from the regression model should drive the main result interpretation and discussion rather than the correlations. In the regression model, it is important that many potential confounding factors, including the sociodemographic characteristics of both parents and children, are controlled.

Response: Thank you for this comment. We acknowledge the limits of relying on correlations while the more advanced statistical test (regression analysis) is available. On the other hand, the study was designed as explorative and the discussion was therefore driven mostly by descriptive results (e.g., the average dislike of parents for various SUPRA, the frequency of various SUPRA). 

However, we have taken your valuable comment into account and have reformulated the relevant paragraph in the discussion section to clarify clear that the parental predictors of SUPRA dislike were identified using linear regression analysis:

“Less positive attitudes towards screen media, higher parental warmth and control and being female (mother) were all found to significantly contribute to high parental dislike of SUR even when analyzed together using regression analysis. Positive parental attitudes towards screen media have been previously studied mostly as a factor contributing to…”

We also replaced the table with the results of regression analysis from the supplement to the main body (Table 4).

In line with this comment, we have removed two paragraphs from the Results section.

1) Second saying that SUPRA was positively associated with daily smartphone time and negatively associated with parental warmth, with a positive correlation between SUPRA and PATS in children but a negative correlation in adolescents.

2) Second saying that Smartphone ownership was the only sociodemographic factor significantly associated with children's SUPRA, with notable differences in usage patterns between owners and nonowners, though no significant differences were found in the total number of routines involving frequent smartphone use.

---------------

Additional changes that were based on / stemmed from the Reviewers’ comments in the manuscript

Thus, we have adjusted the accordingly to the comments:

- Abbreviation SUPRA (smartphone use during personal and relational activities) is now being used throughout the manuscript and in the Tables.

- Title to better fit the content, using the term: relational activities.

- Discussion sect

---

## [Decision Letter · Decision Letter 1]

9 Jul 2024

PONE-D-24-02065R1Parental views on their children’s smartphone use during personal and relational activitiesPLOS ONE

Dear Dr. Gabrhelík,

Thank you for submitting your manuscript to PLOS ONE. After careful consideration, we feel that it has merit but does not fully meet PLOS ONE’s publication criteria as it currently stands. Therefore, we invite you to submit a revised version of the manuscript that addresses the points raised during the review process.

We look forward to receiving your revised manuscript.

Kind regards,

Gal Harpaz, Ph.D.

Academic Editor

PLOS ONE

Journal Requirements:

Additional Editor Comments:

Dear Authors

Thank you for resubmitting the article. It is evident that you have taken the comments of the reviewers into consideration, the article has undergone considerable improvement.

I recommend referring to the specific comments given by reviewer 1 on the latest version, and returning it to the journal as soon as possible.

I would love to see the latest version you submit.

Best regards

Reviewers' comments:

Reviewer's Responses to Questions

**Comments to the Author**

1. If the authors have adequately addressed your comments raised in a previous round of review and you feel that this manuscript is now acceptable for publication, you may indicate that here to bypass the “Comments to the Author” section, enter your conflict of interest statement in the “Confidential to Editor” section, and submit your "Accept" recommendation.

Reviewer #1: All comments have been addressed

Reviewer #2: All comments have been addressed

2. Is the manuscript technically sound, and do the data support the conclusions?

Reviewer #1: Yes

Reviewer #2: Yes

3. Has the statistical analysis been performed appropriately and rigorously? 

Reviewer #1: Yes

Reviewer #2: Yes

4. Have the authors made all data underlying the findings in their manuscript fully available?

Reviewer #1: Yes

Reviewer #2: Yes

5. Is the manuscript presented in an intelligible fashion and written in standard English?

Reviewer #1: Yes

Reviewer #2: Yes

6. Review Comments to the Author

Reviewer #1: I thank the authors for the revised manuscript. The ideas of the manuscript are now more clear, but with this clarification of the main goal of the manuscript, I found the manuscript to be a bit confusing.

In the abstract, you present your goals as follows: "This study aimed to (i) assess parental dislike (PD) of SUPRA, (ii) estimate SUPRA frequency, and (iii) identify predictors of PD of SUPRA"  but on page 7, you put frequency of SUPRA as the last goal, and there is also confusion with the presentation of the result. It would benefit your paper, in terms of being concise and easier to read, to present your goals from a wider range (i.e., frequency of SUPRA) to a specific one (i.e., parental views/dislikes of SUPRA). So I recommend to authors that they think about rearranging the goals of the manuscript in such a way as to get a clearer picture of their study and study goals.

Following that, the discussion of the paper lacks a theoretical explanation of the obtained results. For example, in the beginning of the discussion, you start with "The most negatively perceived was SU during two relational activities: when attending a cultural performance and when a parent is trying to say something important to their child. And additionally, SU during two personal activities: during bedtime and when a child is supposed to focus on something else, such as studying. In contrast, parents were generally okay with two SUPRA: (during waiting and during travelling/commuting), suggesting that SU during these activities may not be seen as intrusive by most parents." However, I haven't seen, throughout the discussion, any explanation as to why you think you got these results. For relational activities, it could be that parents perceive these activities as a chance to bond with children or spend quality time together, and for personal activities, parents may think about the consequences for a child's development, well-being, school achievement, etc. I am certain that there are some theoretical explanations for these and other results you obtained, so I encourage you to include them in your manuscript, as it would give more support to your findings.

I have some other comments and suggestions:

- I recommend including some keywords that reflect your focus on personal and relational activities and maybe excluding the "family" keyword.

- On page 4, in the sentence "This may be due to perceived differences in the purposes of technology use, while daughters are believed to use it predominantly for social networking and sons for gaming [27]" it seems that the word "while" should be exchanged with the word "where".

- On pages 5 and 6, you mention problematic screen media use, but you never define this term.

- Descriptions of Parental warmth and Parental control are missing the used measurement scale.

- "Characteristics" should be included in the method under the Participants section.

- Before the results of the regression analysis, put a heading, such as "Predictors of Parental Dislike of SUPRA" or something similar.

- Have you controlled for other sociodemographic variables in the regression model? Whether the answer is yes or no, you should clarify this in the results (under the regression table), i.e., explain why you have or haven't included other variables.

Overall, the paper shows great improvement after the revision, and I thank the authors for considering the previous comments. The paper describes something still relatively unknown in the literature, and as such, it would benefit from a more theoretical background.

Reviewer #2: The authors have addressed the reviewers’ comments pretty well. Thank you for being so collaborative. I think the paper has become stronger.

7. PLOS authors have the option to publish the peer review history of their article (what does this mean?). If published, this will include your full peer review and any attached files.

Reviewer #1: No

Reviewer #2: No

---

## [Author Response · Author response to Decision Letter 1]

18 Jul 2024

Reviewer #1: I thank the authors for the revised manuscript. The ideas of the manuscript are now more clear, but with this clarification of the main goal of the manuscript, I found the manuscript to be a bit confusing.

In the abstract, you present your goals as follows: "This study aimed to (i) assess parental dislike (PD) of SUPRA, (ii) estimate SUPRA frequency, and (iii) identify predictors of PD of SUPRA" but on page 7, you put frequency of SUPRA as the last goal, and there is also confusion with the presentation of the result. It would benefit your paper, in terms of being concise and easier to read, to present your goals from a wider range (i.e., frequency of SUPRA) to a specific one (i.e., parental views/dislikes of SUPRA). So I recommend to authors that they think about rearranging the goals of the manuscript in such a way as to get a clearer picture of their study and study goals.

Response: Thank you for raising this point. 

We have rearranged the aims in the Abstract as suggested – i.e., from (i) SUPRA frequency to (ii) parental dislike of SUPRA magnitude and (iii) its predictors. We rearranged methods, results and discussion accordingly to mirror this logic.

Reviewer #1: Following that, the discussion of the paper lacks a theoretical explanation of the obtained results. For example, in the beginning of the discussion, you start with "The most negatively perceived was SU during two relational activities: when attending a cultural performance and when a parent is trying to say something important to their child. And additionally, SU during two personal activities: during bedtime and when a child is supposed to focus on something else, such as studying. In contrast, parents were generally okay with two SUPRA: (during waiting and during travelling/commuting), suggesting that SU during these activities may not be seen as intrusive by most parents." However, I haven't seen, throughout the discussion, any explanation as to why you think you got these results. For relational activities, it could be that parents perceive these activities as a chance to bond with children or spend quality time together, and for personal activities, parents may think about the consequences for a child's development, well-being, school achievement, etc. I am certain that there are some theoretical explanations for these and other results you obtained, so I encourage you to include them in your manuscript, as it would give more support to your findings.

Response: Thank you for bringing this to our attention. Given that we have reorganized the order in which we presented aims and reported results, we also reorganized discussion, starting with frequency of SUPRA. Each paragraph on frequency of SUPRA was then supplied with the results on parental dislike of this particular SUPRA, which (as we believe) improved the overall readability of discussion. We included additional sources to better discuss the high dislike of parents for most SUPRA (in total, six new sources on parental attitudes toward screen media were added, mostly reports of qualitative studies). We believe that our discussion of the results has been significantly improved.

Reviewer #1: I have some other comments and suggestions:

- I recommend including some keywords that reflect your focus on personal and relational activities and maybe excluding the "family" keyword.

Response: Thank you, we added “smartphone use during personal activities” and “technoference” and “phubbing” as keywords for smartphone use during relational activites

Reviewer #1: - On page 4, in the sentence "This may be due to perceived differences in the purposes of technology use, while daughters are believed to use it predominantly for social networking and sons for gaming [27]" it seems that the word "while" should be exchanged with the word "where".

Response: Done

Reviewer #1: - On pages 5 and 6, you mention problematic screen media use, but you never define this term.

Response: Done (“...problematic screen media use [33–35], which can be defined as excessive media engagement with impaired control over use and with negative consequences in major areas of life functioning [36]“)

Reviewer #1: - Descriptions of Parental warmth and Parental control are missing the used measurement scale.

Response: Thank you, added.

Reviewer #1: - "Characteristics" should be included in the method under the Participants section.

Response: Included, thank you.

Reviewer #1: - Before the results of the regression analysis, put a heading, such as "Predictors of Parental Dislike of SUPRA" or something similar.

Response: Done accordingly.

Reviewer #1: - Have you controlled for other sociodemographic variables in the regression model? Whether the answer is yes or no, you should clarify this in the results (under the regression table), i.e., explain why you have or haven't included other variables.

Response: We added the information into Methods (“These predictors were selected based on whether they were found to be significantly associated with PD of SUPRA, i.e., on previous step.”) and also when reporting the results: “Using the principle of parsimony, we included only variables significantly associated with the PD of SUPRA in the model.”

Reviewer #1: Overall, the paper shows great improvement after the revision, and I thank the authors for considering the previous comments. The paper describes something still relatively unknown in the literature, and as such, it would benefit from a more theoretical background.

Response: The literature issue has been addressed. Thank you for your time invested in our work.

---

## [Editor Report · Decision Letter 2]

22 Jul 2024

Parental views on their children’s smartphone use during personal and relational activities

PONE-D-24-02065R2

Dear Dr. Gabrhelík,

We’re pleased to inform you that your manuscript has been judged scientifically suitable for publication and will be formally accepted for publication once it meets all outstanding technical requirements.

Kind regards,

Gal Harpaz, Ph.D.

Academic Editor

PLOS ONE

---

## [Editor Report · Acceptance letter]

26 Jul 2024

PONE-D-24-02065R2 

PLOS ONE

Dear Dr. Gabrhelík, 

I'm pleased to inform you that your manuscript has been deemed suitable for publication in PLOS ONE. Congratulations! Your manuscript is now being handed over to our production team.

Kind regards, 

on behalf of

Dr. Gal Harpaz 

Academic Editor

PLOS ONE